# Defining a Cyber Resilience Investment Strategy in an Industrial Internet of Things Context [note 1]

**DOI:** 10.3390/s19010138

**Published:** 2019-01-03

**Authors:** Juan Francisco Carías, Leire Labaka, José María Sarriegi, Josune Hernantes

**Affiliations:** Industrial Management Engineering Department, School of Engineering, TECNUN, University of Navarra, Paseo Manuel de Lardizábal, 13. 20018. San Sebastián, Spain; llabaka@tecnun.es (L.L.); jmsarriegi@tecnun.es (J.M.S.); jhernantes@tecnun.es (J.H.)

**Keywords:** cyber resilience, IoT (Internet of Things), IIoT (Industrial Internet of Things), IT-OT (Information Technology-Operational Technology) convergence, investment policies, system dynamics

## Abstract

The fourth industrial revolution has brought several risks to factories along with its plethora of benefits. The convergence of new technologies, legacy technologies, information technologies and operational technologies in the same network generates a wide attack surface. At the same time, factories need continuous production to meet their customers’ demand, so any stopped production can have harsh effects on a factory’s economy. This makes cyber resilience a key requirement in factories nowadays. However, it is difficult for managers to define effective cyber resilience strategies, especially considering the difficulty of estimating adequate investment in cyber resilience policies before the company has suffered cyber incidents. In this sense, the purpose of this article is to define and model an effective cyber resilience strategy. To achieve this, the system dynamics methodology was followed in order to get five experts’ opinions on the best strategy to invest in cyber resilience. Interviews were conducted with these experts; their reasoning was put into behavior over time graphs and a system dynamics model was built from these findings. The main conclusion is that a cyber resilience investment strategy should be dynamic, investing in both technical security and personnel training, but at first with an emphasis on technical security and later shifting to have an emphasis on training.

## 1. Introduction

Technology today has become crucial in the industrial sector in order to stay competitive. With the paradigm of industry 4.0 or smart manufacturing, companies are now able to be more efficient, more productive and more flexible [1]. The paradigm of smart manufacturing includes a collection of technologies and a convergence of humans, technology, and information. Two of the main technologies it includes are Cyber-Physical Systems (CPS) and the Internet of Things (IoT). Through these technologies, factories aim to obtain real-time data that will let them optimize their decision making, resources and client satisfaction [1,2,3].

With all these advantages, smart manufacturing has also brought a convergence between the Information Technologies (IT) and the Operational Technologies (OT) [3,4,5]. This convergence has increased the companies’ risk and insecurity since OT systems have unique characteristics that cannot be easily protected with standard IT security practices [6]. Even company directors are aware that their risks are greater as technology is more embedded in their processes, but at the same time feel that companies are not prepared to face the challenges and implications of these new risks [3,5,7].

In fact, cyber incidents can have severe impacts on companies. The economic damage after a cyber incident comes from different sources: loss of reputation and clients’ trust, stopped production or services, loss of intellectual property, fines and contractual payment obligations, etc. [8,9,10,11]. In fact, several reports show that cyberattacks are one of the main risks for companies these days [12,13] and that cyber incidents cost around $445 billion annually [13].

In the industrial context, probably the most cited example of a cyber incident is Stuxnet, a malware designed to target the industrial control systems (ICS) and slow down the uranium enrichment plants outside of Natanz (Iran) in order to limit Iran’s nuclear weapon production in 2009. To do this, the malware made the centrifuges of the uranium enrichment plant spin faster than they were supposed to, and then slower to break the already fragile machines. The exact number of broken centrifuges in Natanz is unknown, but authors estimate between 900 and 2000 centrifuges were broken in less than a year [14]. 

Although Stuxnet was an attack on nuclear plants and many consider it to be the first announcement of cyberwarfare [14,15], it was a direct attack on an ICS and similar attacks could affect any kind of connected factory.

In this sense, in order for companies to thrive in this new industrial era, they must broaden their cybersecurity concept from only perimeter security to a prevention, detection, response and recovery point of view [16,17]. Many important institutions and researchers use the concept of cyber resilience to describe this broader point of view [18,19,20,21]. However, building cyber resilience is not a merely technical problem and it cannot be solved just through technological security tools. In fact, many cybersecurity problems in companies are caused by humans and the vulnerability that this human factor brings cannot be underestimated [7,18]. This means that for a company to be cyber resilient it needs to avoid focusing only on perimeter security [6], it needs to be prepared to face the before, during and after periods of an attack with both factors, technology and people, in mind [18,22].

Even though managers are becoming aware of the importance of including cyber resilience policies, there is still a significant number of companies that lack adequate funding for security practices [7]. Besides, it could be overwhelming to budget and execute a cyber resilience strategy because of the multiple aspects it encompasses, such as protective technology, awareness and training, asset management, response and recovery planning, etc. [23]. Moreover, it is difficult for managers to define an effective cyber resilience strategy considering that most of the time they lack a sufficient budget [7,24] and it is difficult to correctly budget for cyber resilience when there has not been any cyber incident experience yet to indicate the cost and effects it has. Besides, since the budget allocation can also change depending on the types of attacks that the company could expect, budgeting is different to for a company that expects only opportunistic attacks than for a company that can be targeted for an attack [25], adding another difficulty to the definition of an effective cyber resilience strategy.

The current literature seeks to optimize the investment in security measures for companies, however, most of them focus only on technological investments and how to find the balance between minimum investment and enough protection [24,26,27,28,29,30]. For this reason, these studies use game theoretic models, optimization models or a combination of both, but neither kind of model considers the variability and risks that the humans would include in a real situation. In other words, most of the security investment is done in anti-viruses, firewalls, encryption, intrusion detection systems, data backup, and hardware devices; but not in the human aspect that is often targeted through social engineering [30]. These studies also approach the optimization of a cyber resilience investment strategy with an economic point of view, however estimating cyber resilience costs is often a problem on its own, with its own implications [31].

The purpose of this article is to address the above-mentioned difficulties through the analysis and modeling of an effective cyber resilience investment strategy based on several experts’ mental models (i.e., ideas) of how a connected factory would behave in different scenarios. This approach is different from previous studies because it uses a cyber resilience approach where the investment in security is not limited to technological measures and it uses a system dynamics approach where it is possible to model soft variables for which quantitative data is not available, but in practice play important roles in the behavior of other variables [32]. This approach is not focused on getting hard data on the economic part of the cyber resilience strategy, but on getting the knowledge and experience embedded in the experts’ minds.

Nowadays, this kind of study is especially important since the rapid evolution of technology and the lack of awareness from factory managers puts smart factories at high risk [5,7,33] and, as mentioned above, makes cyberattacks one of the risks with a higher impact and higher probability of occurring [12,13].

To achieve this article’s objectives, several interviews have been carried out with different cybersecurity experts to obtain the behavior over time of key variables in a hypothetical industrial company that has implemented the smart manufacturing paradigm. These behaviors were modeled and simulated in a system dynamics model for a better understanding of the experts’ reasoning and to check for unexpected long-term side effects of the reasoning they proposed in a dynamic simulation. The development of this study faced different challenges such as: defining a suitable scenario that all experts would understand equally considering their different profiles, defining the key variables that should be included in the scenarios the experts would analyze, and reaching a consensus on the behaviors of these variables. The latter was the most difficult to overcome, and thus the methodology is prepared to have several rounds of proposals until consensus was reached.

The article is structured in such way that: Section 2 of this article explains in a detailed manner how the interviews were done, and how the model was constructed. Section 3 shows the results of the interviews with the experts, the hypothesis drawn from these graphs, the model that was created and the simulations done with the model. Section 4 is a discussion of the results where the experts’ graphs are contrasted to the model simulations’ graphs. Finally, Section 5 summarizes the conclusions the authors of this article have drawn from this article.

## 2. Methodology

As stated in reference [34], the system dynamics (SD) modeling and simulation methodology is suitable for the purposes of this article. SD has been used in several disciplines of research such as in engineering, scientific humanitarian sciences, the economy, manufacturing and management, planning and logistics, healthcare, urban planning, etc. The SD simulation and modeling methodology uses a top-down approach that allows us to manage and analyze complex adaptive systems involving interdependencies [35,36]. This methodology is grounded in the theory of nonlinear dynamics and feedback control, which deals with the internal feedback loops and time delays that influence the whole system [37]. 

SD methodology allows the effective modeling of socio-technical systems, consisting of human, organizational and technological parts. This is possible because the SD methodology allows models to deal with “soft” variables that cannot be quantified, but that are often important factors that influence the behavior of other variables in the problem that is being modeled [32].

Another characteristic of SD is that it is used when the individual properties are not decisive and high-level aggregation is desired or required for management purposes. This is typically the case for management strategies and long-term planning [38,39].

Based on these specific characteristics of this modeling methodology, SD is especially suitable for the analysis of cyber resilience management. On the one hand, cyber resilience development involves multiple relationships between variables and the complexity increases because of the presence of significant delays. The process of building cyber resilience in an organization needs to consider variables that evolve quickly, like new types of cyber-attacks or software upgrades, with others that need longer times to change, such as organizational culture or individual attitudes towards security. On the other hand, building cyber resilience involves “soft” variables that are hard to measure, such as “employees’ training level”.

To follow the methodology, the first step is to find the reference behavior modes that describe the problem to be modeled [40]. The reference behavior modes in the context of SD modeling refer to the behaviors over time (BoT) of the key variables in the problem. It is important to notice that, due to the nature of the variables involved in cyber resilience management, it is often difficult to find quantitative data on these variables. However, experts can graph these variables because they have acquired that knowledge through experience. By acquiring these experts’ experience and graphing it in a pseudo-quantitative way, the results can be used as input for a model to represent reality. Though this is not hard quantitative data it is vital to understanding the dynamics of managing cyber resilience [32,34].

For this reason, in order to find the reference behavior modes for an industrial factory implementing IoT devices, five experts were individually interviewed. The experts’ profiles were: two cybersecurity researchers that work closely with industrial companies, one CISO from a company, and two software engineers that work for industrial companies. All of these experts have vast experience in the industrial context, its current evolution, and the trade-offs of managing cyber resilience in a company.

During the interviews, the experts were asked to draw the BoT graphs of four variables (training level, technical security level, management’s awareness, and cyberattacks’ success probability) in three scenarios that only differed in the investment policies. The three considered scenarios were: (1) equal investment in personnel training and technical security, (2) significantly more investment in personnel training than in technical security, and, finally, (3) significantly more investment in technical security than in personnel training.

The scenarios were 250 weeks (around 5 years) of a hypothetical industrial company that started to invest in IoT and as a result, by week 20, had multiplied the number of connected devices to the network by five, abruptly increasing their attack surface. As a simplification, they were told that the change was immediate (such as the graph shown in Figure 1) and that before week 20 the company had a reasonable technical and personnel training level; meaning that they could assume that before that week the situation was stable for all of the variables.

To make sure all of the experts understood the same for the presented variables, the following definitions were given to them:**Cyberattacks’ success probability**: variable that shows how likely is a cyberattack to succeed in the current company’s conditions. As a probability, this variable is dimensionless (does not have units) and has a range from 0 to 1.**Training Level:** this variable shows how much knowledge the non-IT personnel has about the existence of threats to the current systems, their role in preventing them, and what to do in case they detect one. In other words, it measures the ability of the non-IT personnel to follow the established policies to prevent, detect, resist and recover from vulnerabilities and cyberattacks. The training level will also reflect the ability of the personnel to improvise and adapt to crisis situations or incidents whose solution is not previously known. Since this is a soft variable it is dimensionless and in this case, is considered to be proportional to the threats the company currently faces.**Technical security Level:** measures the ability of the installed technology to prevent, detect, resist and respond to the company’s current threats. In other words, how protected the company is from external threats through their antivirus software, firewalls, Intrusion Detection Systems (IDS), software security updates, etc. Like the training level, this variable is dimensionless and proportional to the threats the company currently faces.**Management’s awareness Level:** shows the knowledge and commitment of the management about the threats that the company faces and their potential impacts. The higher the management’s awareness the more likely they are to invest in technical security or personnel training. Since this is also a soft variable it is dimensionless as well.This variable could be influenced by the personnel’s training (since the personnel includes the management) or by the concern owed to the effects of a suffered cyber incident.

Once the experts had drawn the BoT graphs of these variables for the three scenarios they were asked several questions to see what would change if there were two cyber incidents in the company, one in week 90 and the other one in week 150. The questions emphasized focusing on two main aspects, the values of the variables they had already drawn and possible differences in the second incident once the company had suffered the first one.

The main goal of the five interviews with the experts was to capture their mental models and the reasoning behind them. With all this information, the experts’ mental models were compiled into one set of BoT graphs and textual descriptions about the reasons for these behaviors. These BoT graphs were sent with a written explanation to the experts to see if they agreed with those behaviors. When an expert disagreed with the graphs or their explanation, these were adapted to the expert’s suggestions and the process was repeated until consensus was achieved.

After getting the reference behavior modes, the SD methodology requires a dynamic hypothesis [40]. In SD the dynamic hypothesis is usually represented in a causal loop diagram (CLD), which represents the causal relationships between variables, the feedback loops they produce, and the interactions between feedback loops. For this article, with the BoT graphs the experts drew and their reasoning, the CLD was developed.

Finally, with the CLD defining the interrelationships of variables in the problem to be modeled, an SD model was developed. The model was designed to apply the experts’ reasoning, this way a better understanding of their point of view could be achieved, their reasoning could be checked for unintended side-effects in dynamic simulations and conclusions could be drawn.

## 3. Results

Using the methodology described above, three main results have been obtained. These three results are the reference behavior modes, the dynamic hypothesis, and the SD model that represents this behavior. The following subsections present these results in a detailed manner.

### 3.1. Reference Behavior Modes

After the interviews with the experts and the subsequent consensus-seeking process described previously, the BoT graphs obtained and their explanations were the following.

#### 3.1.1. BoT for an Equal Investment Scenario

For a scenario with equal investment in technical security and in personnel training, the experts consider the BoT would look like the one in Figure 2.

As shown in the graph, in a scenario with equal investment in technical security and in training, experts agree that the training level and the technical security that the company had before the investment in IoT technology is not enough after multiplying the number of devices in the network by five. The reason for this is that the personnel does not have enough knowledge about the new devices nor the risks that they bring to the company and that the new technical vulnerabilities are probably not covered with the currently installed technology. Therefore, there is a sudden decrease in the technical security and training levels at week 20 when all the new devices are installed.

Even though the investment in technical security and training is the same in this scenario, experts agree that the learning process is slower than the technology installation process, which they believe has an almost immediate effect. For this reason, the training level has a higher time delay than security technology in the BoT graph for this scenario. As a consequence, training reaches reasonable levels (like before week 20) later in time than technical security.

At the same time, since there is not enough training and technical security in the company for those new devices, the success probability of a cyberattack acutely increases during the period where these variables are not yet at adequate levels. However, as the company starts implementing more technical security and the personnel is trained (after week 20), the probability starts to decrease.

Meanwhile, the management’s awareness increases slowly between weeks 20 to 90 just due to the training they receive since it lets them have a better understanding of the threats the company is exposed to.

On week 90, when the experts were told that the company suffered the first cybersecurity incident, the management’s awareness noticeably increases for a couple of weeks because of the sudden concern that the incident brought to them. Though the peak awareness is short-lived, the management’s awareness level stays higher than before the company suffered the incident, because the experts agree that the management learns from incident to incident. 

Also, according to the experts, the management would change its investment policies in order to mitigate the cause of the cyber incident. Meaning that, if the cause was a human mistake, they would invest more in training, but if the cause was a technological gap, they would invest more in technical security. In this case, the cause of the cyber incident was not specified in the scenario, so experts thought that it was safe to assume that the failure would be caused by each case’s lowest-valued variable between the technical security level and the training level, or in other words their “weakest link”. In this scenario, at week 90 the training level had a lower value than technology, so the experts assumed that the cause of the incident was a human mistake and that the management would invest more in training after the incident. Hence, the training level temporarily increases more than the technical security level after week 90. This effect, however, is not so immediate because of the delay between the investment and the increase in training level.

Experts have also considered that the success probability of a cyberattack increases drastically after a cyber incident. The cause of this is that the probability becomes a conditional probability, and their experience shows that it is usually easier to have a successful cyberattack right after another attack. However, after the incident and its causes are mitigated through the training and technology in the company, this probability starts to decrease again.

Finally, similar effects to those after week 90, can be observed after the second cyber incident at week 150. However, in this case, the increases in the technical security and training levels are smaller, because experts agree that after achieving a certain level in both variables it becomes harder and more expensive to increase these levels. This certain level of technical security or training where it becomes harder and more expensive to increase will be further referred to as the saturation level.

It is also worth mentioning that experts agree that this scenario would have the lowest cyberattack success probability of the three scenarios towards the end of the simulation. The reason for this lower success probability, according to the experts, is that neither technical security nor training can work without the other, and the mix of both of them will make the company more resilient in the long term.

#### 3.1.2. BoT for a Training Biased Scenario

In a scenario where the company invests significantly more in training than in technical security, the experts consider the BoT would look like the graph in Figure 3.

In this scenario, experts have agreed that at week 20 the decreases in technical security and training levels would be similar to the previous scenario. The reason for this decrease is also the same: the cybersecurity technology and the training that the personnel had before is not enough for this sudden change.

However, when these variables start increasing because of the investment (after week 20), the training level increases faster than in the previous scenario and technical security level would reach a reasonable level (similar to the one before week 20) later in time due to the defined scenario.

As a consequence of the increased investment in training, the management reaches higher awareness levels in less time. According to the experts, the increase in training investment would also cause the awareness levels to stay at peak after-attack levels for longer than in the other two scenarios. This is because the training they receive lets them understand the risks the company faces better and thus be more aware. 

Also, as in the previous scenario, the cyberattacks’ success probability would increase steeply after week 20 and decrease as the technical security and training levels increase. There would also be a temporary increase in this probability after the cyber incidents at weeks 90 and 150 as well.

The behavior of the training and technical security levels after the cyber incidents is similar to the previous scenario. Meaning that even though the investment in both variables increases because of the management’s concern, as in the previous scenario, experts consider there will be a change in the investment policies that will cause a higher increase in the “weakest link”. In this case, the technical security level is lower than the training level; hence, the experts have assumed that the cause of the incident was technological and therefore, they think there would be a higher investment in technical security and as a consequence, there is a higher increase in this variable.

Experts agree that the success probability of a cyberattack in this scenario stays the highest of the three scenarios throughout the whole simulation. They agree that even though both variables, the human and the technological, are important, technology reduces the cyberattacks’ success probability the most. The reason behind this is that industrial IoT technology itself brings technological vulnerabilities that cannot be controlled by humans, and that there are several exploits and attack vectors that cannot be mitigated only by humans, no matter how trained they are.

#### 3.1.3. BoT for a Technical Security Biased Scenario

Figure 4 shows how the experts consider the key variable’s BoT would be in a scenario where the company invests significantly more in technical security than in training.

On a third and last scenario where there is more investment in technical security than in training, the decrease in training and technical security levels at week 20 maintains, but, when both start to increase due to the current investment policies, technical security is the one that reaches saturation faster. Experts agree that saturation in technology would be even faster than the saturation in training in the previous scenario and that due to its delay, the training level would stay even lower than the technical security level in the previous scenario.

As mentioned in the previous scenario, the technical security level is very important for the cyberattacks’ success probability, hence this variable decreases faster in this scenario than in both of the previous ones. However, in the long term, the lower training level leaves the cyberattacks’ probability of success higher than in the first scenario.

The management’s awareness would grow the slowest in this scenario, because of the lower investment in training. This results in a lower overall curve than in the previous scenarios. 

In this scenario, the behaviors after the cyber incidents are similar to the previous scenarios. In this case, the management would invest more in training because, according to the experts, it is most likely the cause of the cyber incident since it has a lower value than the technical security level. Therefore, after the time delay that this variable has, it will increase more noticeably than the technical security level after both incidents.

### 3.2. Dynamic Hypothesis

From the BoT graphs and the reasoning that the experts have agreed on, a dynamic hypothesis with four feedback loops is proposed. These four feedback loops are depicted in the CLD shown in Figure 5.

The arrows in the CLD represent causal relationships, and the (+) sign near the head of the arrows between variables means that the causal relationships are directly proportional. On the other hand, the (−) sign represents inversely proportional causal relationships. 

The causal relationships in the CLD form feedback loops that can be of two types: reinforcing or balancing. Reinforcing loops represent behaviors where there is exponential growth or decay due to the interaction between variables involved in them. Balancing loops, on the other hand, represent limiting behaviors.

In this case, the dynamic hypothesis, through its four feedback loops, describes four main behaviors: Training reinforces awareness (loop R1).Cyber incidents can increase the chances of having another cyber incident (loop R2).The management invests in training to improve cyber resilience (loop B1).The management invests in technical security to improve cyber resilience (loop B2).

Besides the four main behaviors mentioned above, notice that the CLD considers the fact that investing in connected devices generates more attack surface and thus a higher cyberattacks’ success probability. Although it is not inside a feedback loop, this consideration is important because it represents the real scenario companies are living and therefore, it is key to represent the problem that is being modeled.

It is also important to mention that not all the feedback loops in this hypothesis occur with the same frequency. B1, B2, and R2 can only act completely after a cyber incident occurs. This represents the fact that managers will become more aware when there is a cyber incident (B1 and B2), and that cyber incidents increase the probability of success of other cyberattacks due to the compromised systems (R2).

### 3.3. SD Model

SD models are usually represented with a stocks and flows diagram. This kind of diagram can depict a system in a way that the variables involved, and the relationships between them can be translated into mathematical formulas. 

In a stocks and flows diagram, stocks represent the accumulation of material or state variables and are portrayed as rectangles. On the other hand, flows are represented as valves that control the inflow or outflow of the stocks. 

Other variables involved that are neither stocks nor flows are called auxiliary variables. This kind of variable is necessary in order to close causal relationships inside any system. Moreover, the arrows that connect variables in a stocks and flows diagram represent the aforementioned causal relationships so that if variable A has an arrow that comes from variable B, then variable B is a cause of variable A and thus it will appear as part of the mathematical formula for variable A. Notice that variables A and B in an SD model can be stocks, flows or auxiliary.

One stocks and flows diagram that models the dynamic hypothesis posited in the previous subsection is shown in Figure 6.

As mentioned in Section 3.2, the dynamic hypothesis was made based on the experts’ reasoning, therefore the simulations done with the same scenarios that the experts were given should result in similar behaviors as the BoT that were explained in Section 3.1.

Figure 7 shows the BoT generated in a simulation of this model in an equal investment scenario. To interpret this and the following BoT graphs take into account that due to the fact that the variables in this model are soft or dimensionless, no numerical scale, even not an arbitrary one, has been provided. Accordingly, the values on *y*-axes should be interpreted as expressing low, medium or high values. In this sense, the features of interest in the graphs that should be taken into account instead are changes in shapes, inflections, maxima and minima.

In a training biased scenario, the BoT graph generated by the model looks as presented in Figure 8, which can be compared to Figure 3.

Finally, in a technical security biased scenario, the BoT graphs generated by the model look as presented in Figure 9, which should be compared to Figure 4.

Although the graphs generated by the model look fairly similar to the ones drawn by the authors, there is one difference that should be emphasized: as Figure 8 and Figure 9 show that in a training biased scenario, the technical security level reaches higher levels faster than in a technical security biased scenario. This behavior, despite being paradoxical, has an explanation that will be further discussed in the next section.

## 4. Discussion

The literature posits that there are many domains to be considered when building cyber resilience in a company. These domains have a very heterogeneous nature as they include internal, external, technological, cultural and human aspects [18,19,22,23]. This heterogeneity makes building cyber resilience a complex process where managers have to invest time and money in different cyber resilience policies that encompass these domains. Even though this is widely known, there is no roadmap for the order in which these policies should be implemented or prioritized. In this line, this article’s results show the following:

After interviewing experts using hypothetical scenarios with different investment policies it is clear that technology plays an important role in the cyber resilience building process. In fact, as explained in previous sections, experts believe technology is the most important aspect to invest in when building cyber resilience since it is the one that will most significantly reduce a cyberattacks’ success probability and prevent it from becoming a cyber incident.

Nonetheless, experts have also agreed that training the personnel is still important in the cyber resilience building process. They unanimously agreed that neither technical security nor personnel training could build cyber resilience completely without the other being present.

Therefore, according to the experts, an investment should be made in both technical security and personnel training. However, taking into account their reasoning the investment strategy should be dynamic and change from technical security-biased policies to training-biased policies once the technology is saturated. Notice that this does not mean that companies should not invest in training until they have saturated the technical security level, but that without neglecting training they should focus on technical security at the beginning, and when they have a reasonable technical security focus on improving training. This way the company would minimize the cyberattacks’ success probability faster.

For a further understanding of the experts’ point of view, it was important to input their reasoning into a model. Even though no model can completely depict reality, some can be really useful to draw conclusions or show unexpected results that could affect a system. In fact, a distinctive feature of SD models is their ability to capture these unintended side-effects and to help identify policies to minimize their impact. This is crucial since experience teaches that many strategies that initially seem to work often have disastrous unintended long-term consequences [35,36,41].

In this case, the model shows one unexpected behavior that could happen when applying the experts’ strategies to reality. The unexpected behavior is that in an equal investment scenario and in a training biased scenario, the company might end up investing more in technical security than in a technical security biased scenario. This is due to the dynamic behavior caused by the interaction between R1 and B2 feedback loops. The increase in the training level increases the management’s awareness level (R1), and the increase in management’s awareness level increases the investment in technical security (B2).

Even though when simulated and reasoned the results seem to be coherent with the experts’ reasoning, they were interviewed once more to see the results of the simulations and asked if they agreed with those behaviors. In these interviews, the unexpected behavior was explained to the experts with the graphs generated in the simulation and consensus BoT graphs were shown to them at the same time so they could see the difference more clearly.

In these interviews, the experts agreed that the unexpected behavior was reasonable, especially since in the scenario with a technical security bias, before the first cyber incident, the company is roughly at the same level than before installing the IoT devices. The experts reasoned that the behavior shown in the simulations explained the following: In a scenario where the management’s awareness does not increase, even when their strategy is to invest as much as possible in technical security, they settle for having a technical security level similar to the one before the increase in IoT. According to them, the management in a company such as this one would only invest to have more technical security when they have been affected by the cyber incident and thus are more concerned.

After the interviews, experts also acknowledged that there is a bias in the industry that favors technical security over personnel training, even though it is a commonly cited fact that the “weakest link” in cybersecurity is the human error.

It is important to highlight that the model emphasizes the effect of training the personnel, and especially the management, showing that this cannot be underestimated in a company; but that this result does not mean that the experts were wrong, or that the best strategy is to invest equally in personnel training and technical security from the beginning. On the other hand, this result stresses that even though when starting to invest technology should be a priority, training should not be completely forgotten, and that, as the experts have said, the technical security biased strategy should at some point be changed to a training biased strategy for an optimal strategy.

It is also important to highlight that the model developed in this paper, besides helping simulate and better understand the experts’ ideas and experience, could be a useful tool to raise awareness among factory managers because it depicts in simple graphs the behaviors of the main variables whose values are not easily quantifiable (soft variables). Knowing these behaviors and how the variables in cyber resilience management interact could, by itself, lead to a better and more effective management strategy. Besides, the model also depicts how these variables would evolve/change in situations such as: rapidly increasing the number of IoT devices and suffering a cyber incident in a highly connected factory; so, through the different simulations this model can help factory managers develop strategies to react to these situations. Thus, this model could be a useful tool in the decision-making process of a factory manager.

## 5. Conclusions

This article has found a roadmap to an efficient investment strategy for building cyber resilience. In order to achieve this, experts were interviewed and their points of view were summarized into behavior over time graphs of several key variables that led to the modeling of the problem in a system dynamics model.

The experts’ point of view and the system dynamics model led to the conclusion that in the cyber resilience building process, both technology and personnel training are important, and neither should be overlooked in an investment strategy. However, to efficiently invest in cyber resilience the first step should be to invest more in tools and technical solutions, and when these are in place change to a more training-centered investment. This strategy will help factories minimize any cyberattacks’ success probability faster and more efficiently.

The model developed in this paper can be used as a tool in the decision-making process of factory managers since it raises awareness and depicts in simple graphs how the important, yet intangible, variables of cyber resilience management behave in situations such as the rapid increase of IoT devices and the suffering of a cyber incident in different investment scenarios.

## Figures and Tables

**Figure 1 sensors-19-00138-f001:**
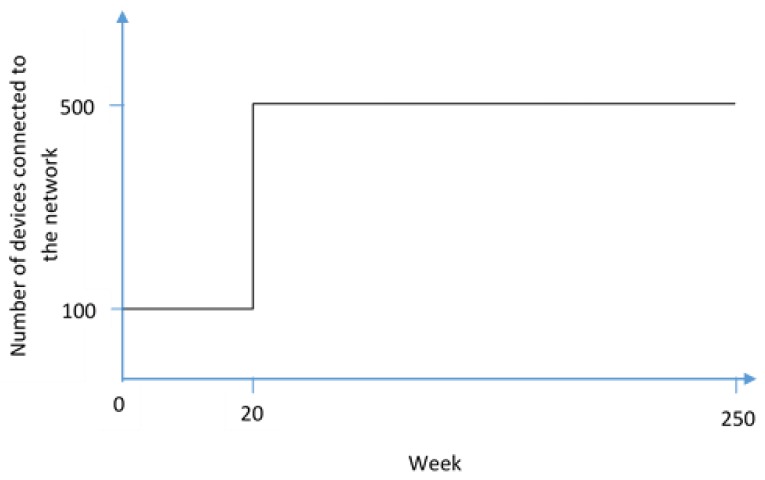
Change in devices connected to the network at week 20.

**Figure 2 sensors-19-00138-f002:**
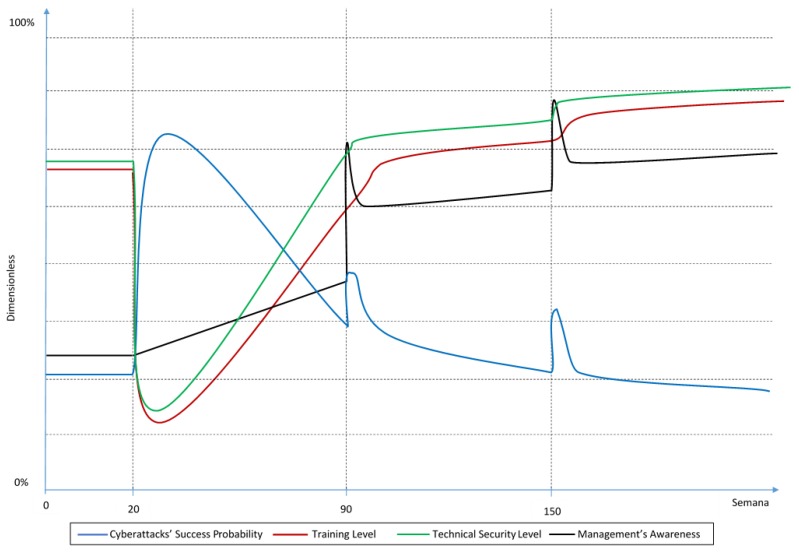
Behavior over Time (BoT) for an equal investment scenario.

**Figure 3 sensors-19-00138-f003:**
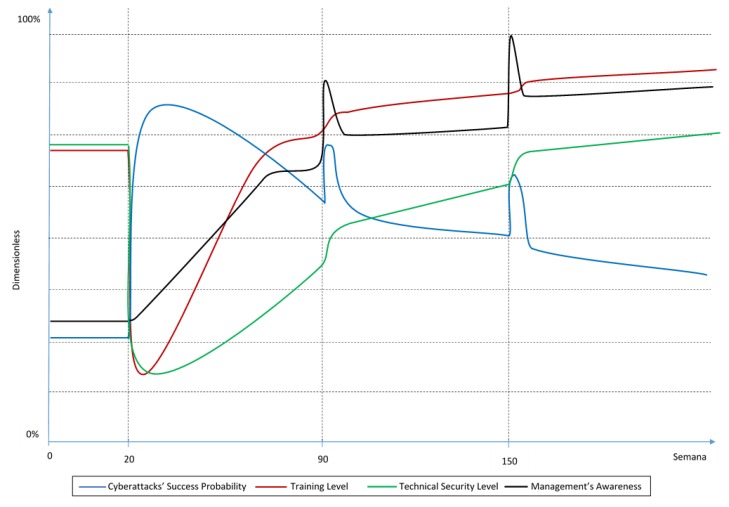
BoT for a training biased scenario.

**Figure 4 sensors-19-00138-f004:**
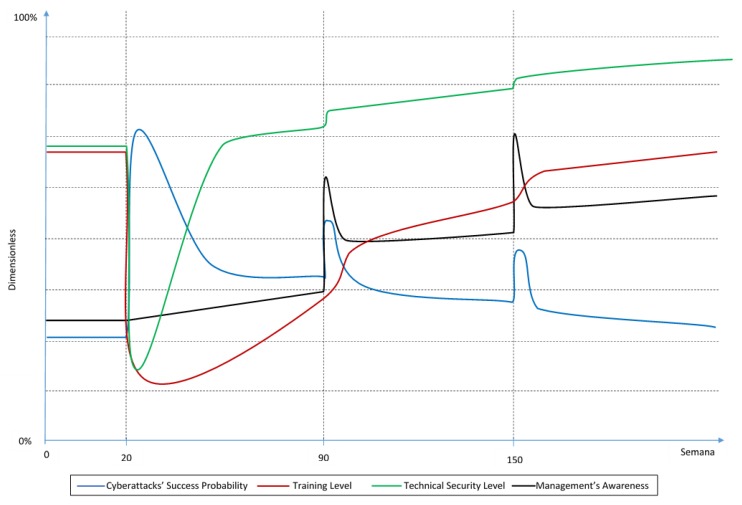
BoT in a technical security biased scenario.

**Figure 5 sensors-19-00138-f005:**
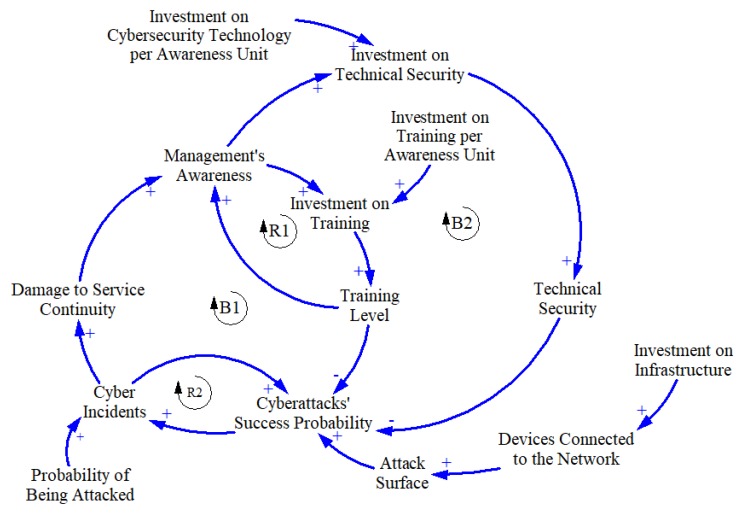
Causal loop diagram.

**Figure 6 sensors-19-00138-f006:**
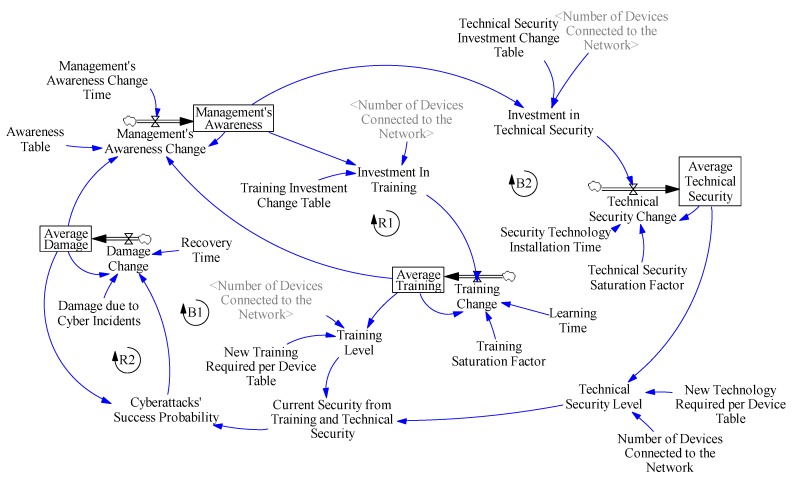
Stocks and flows diagram.

**Figure 7 sensors-19-00138-f007:**
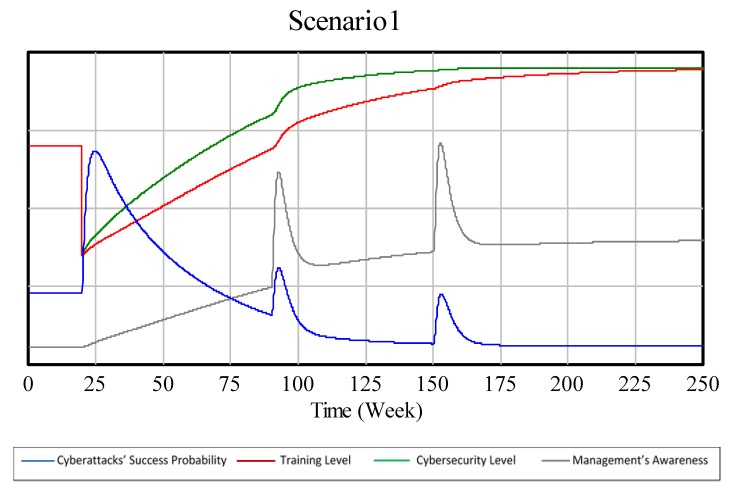
SD model’s BoT of an equal investment scenario (compare to Figure 1).

**Figure 8 sensors-19-00138-f008:**
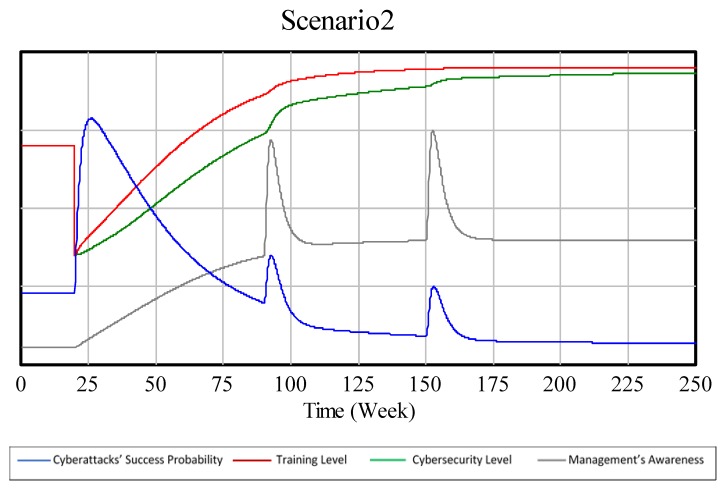
SD model’s BoT of a training biased scenario.

**Figure 9 sensors-19-00138-f009:**
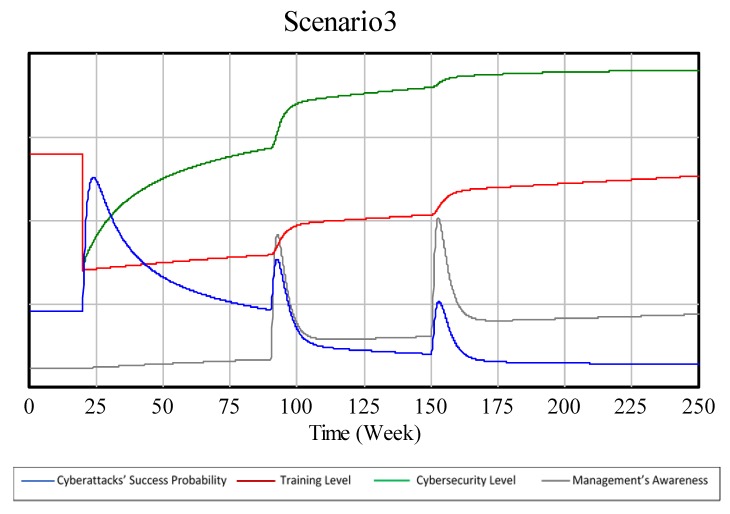
SD model’s BoT of a technical security biased scenario.

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
