# Peer review of "Defining a Cyber Resilience Investment Strategy in an Industrial Internet of Things Context†"

_sensors, 2019, doi:10.3390/s19010138_

Round 1
Reviewer 1 Report
This paper defines and models an effective cyber resilience strategy. To achieve the objective, the authors follows the system dynamic methodology to get experts; opinion. The conclusion of this paper is that a cyber resilience investment strategy should be dynamic, investing in both technical security and personnel training, but at first with an emphasis on technical security and later change to have an emphasis on training. The following are the suggestions. The authors may consider to revise the paper accordingly. - What is the motivation of this paper? The authors may consider to revise the Introduction section with a concrete motivation. - What are the challenges faced by the authors while conducting this research? - Why this research is important? What is the difficult part? - Is there any related or similar studies around this topic? How the existing studies different from the current proposal.Author Response
Dear reviewer 1,
We thank you very much for your review on our article, which has been very helpful to improve our research. In order to address your comments, we have added and highlighted in yellow some parts in the article. Going point by point through your comments, we have done the following changes:
1. What is the motivation of this paper? The authors may consider to revise the Introduction section with a concrete motivation.
To address this comment we have added the following text to explain the gap we are trying to fulfil, and, in addition, make the purpose of the article and its main objective explicit in the introduction section:
· “The current literature seeks to optimize the investment in security measures for companies, however, most of them focus only on technological investments and how to find the balance between minimum investment and enough protection [24,26–30]. For this reason, these studies use game theoretic models, optimization models or a combination of both, but neither kind of model considers the variability and risks that the humans would include in a real situation. In other words, most of the security investment is done in anti-viruses, firewalls, encryption, intrusion detection systems, data backup, and hardware devices, and not in the human aspect that is often targeted through social engineering [30]. These studies also approach the optimization of a cyber resilience investment strategy with an economic point of view, however estimating cyber resilience costs is often a problem of its own, with its own implications [31].”
· “The purpose of this article is to address the above mentioned difficulties through the analysis and modeling of an effective cyber resilience investment strategy based on several experts’ mental models (i.e. ideas) of how a connected factory would behave in different scenarios.”
2. What are the challenges faced by the authors while conducting this research? What is the difficult part?
To address these questions, we have added the most important challenges we faced during the research in the following sentence: “The development of this study faced different challenges such as: defining a suitable scenario that all experts would understand equally considering their different profiles, defining the key variables that should be included in the scenarios the experts would analyze, and reaching a consensus on the behaviors of these variables. The latter was the most difficult to overcome, and thus the methodology is prepared to have several rounds of proposals until consensus was reached.”
3. Why this research is important?
To address this comment we have added the following sentence explaining the importance of our work: “Nowadays, this kind of study is especially important since the rapid evolution of technology and the lack of awareness from factory managers puts smart factories at high risks and, as mentioned above, makes cyberattacks one of the risks with higher impact and higher probability of occurring [12,13].”
4. Is there any related or similar studies around this topic? How the existing studies different from the current proposal.
Trying to give more information on the gap to support our motivation for this paper we added more references and addressed what is being studied in similar research. This corresponds with the first bullet point in the first comment’s response.
We have also explicitly added the most important differences between our approach and the ones used currently in the literature in the following sentence:
· “This approach is different from previous studies because it uses a cyber resilience approach where the investment in security is not limited to technological measures and it uses a system dynamics approach where it is possible to model soft variables for which quantitative data is not available, but in practice play important roles in the behavior of other variables [32]. This approach is not focused on getting hard data on the economic part of the cyber resilience strategy, but on getting the knowledge and experience embedded in the experts’ minds.”
We hope that you see these changes fit to answer your questions and help make our paper clearer. If not, we would be glad to receive more feedback on our paper and will be open to further improving it.
Again, thank you very much for your feedback.
Kind Regards,
The Authors.
Reviewer 2 Report
The article presents an action plan for an effective investment strategy for building cyber resilience. Interviews with experts were made and their points of view were summarized. The authors concluded that in the process of building cybernetic resilience, training and increasing investment in tools and technical solutions are important.
Research raises doubts on the basis of information from experts. No hard information about the reliability and quality of this data. Models have been developed based on information from experts? The results are actually obvious. It also seems to me that Sensors is not an appropriate magazine for this type of article.
Author Response
Dear Reviewer 2,
We deeply appreciate your feedback on our research. We are sure it has added value to our article and helps us improve our current and future research. We humbly accept your comments and respond to them, point by point, in the following way:
1. Research raises doubts on the basis of information from experts. No hard information about the reliability and quality of this data.
To address this comment we have added more information on the background of the experts and we have emphasized and referenced the importance of “soft” variables that are not easily quantifiable, but are key in the dynamics of some systems such as the one being modeled. This can be seen in the following sentences:
· “It is important to notice that, due to the nature of the variables involved in cyber resilience management, it is often difficult to find quantitative data on them. However, experts can graph these variables because they have acquired that knowledge through experience. By acquiring these experts’ experience and graphing it in a pseudo-quantitative way the results be used as input for a model to represent reality. Though this is not hard quantitative data it is vital to understand the dynamics of managing cyber resilience [32,34].”
· “All of these experts currently work in projects that require them to manage cyber resilience and have vast experience with this kind of project, the industrial context and its current evolution, and the trade-offs of managing cyber resilience in a company.”
Also, as mentioned in our paper, we are aware that our model, like every quantitative or qualitative model, cannot completely depict reality. However, it can be useful as it has valuable information summarized in simple graphs about variables that are hard to measure but whose behaviors are embedded in the experts’ minds due to their experience and knowledge of the field. Thus, we truly believe that, even though the experts’ opinions are not and will never be the complete picture, this kind of model can still be a reliable source of information and an especially useful one.
2. Models have been developed based on information from experts?
It is correct. The model we have developed and used for the simulations is based on the information gathered from the experts. As explained in the methodology of the article, the experts’ graphs are considered as the reference behavior modes and are the base of the dynamic hypothesis used to develop the model.
3. The results are actually obvious.
Though the results presented in the paper may seem obvious once they are explained, during our research it was not as easy to reach a consensus among experts with such heterogeneous profiles. In this sense, we have added the following sentences where we remark some of the challenges we had, and highlight that reaching consensus was one of them.
· “The development of this study faced different challenges such as: defining a suitable scenario that all experts would understand equally considering their different profiles, defining the key variables that should be included in the scenarios the experts would analyze, and reaching a consensus on the behaviors of these variables. The latter was the most difficult to overcome, and thus the methodology is prepared to have several rounds of proposals until consensus was reached.”
Also, as seen in the results and discussion sections, even after consensus was reached, there were behaviors that the experts could not foresee.
4. It also seems to me that Sensors is not an appropriate magazine for this type of article.
This paper is submitted to this special issue due to an invitation to extend our conference paper from GIoTS 2018. We are aware that it is not the approach that a typical Sensor’s article would use. However, we believe that a paper like ours, though not as technical or technology-centered as most of Sensor’s articles, can still fit in the scope of the journal. The reason behind this is that technology and technological evolution can be closely related to the risk of a company, especially an industrial one. In this sense, we think that it is important to do research on the technology management as well as in technology itself. Therefore, we humbly believe that a paper like ours could be a very helpful complement to the typical Sensors article.
We hope the changes made to the paper have answered your comments. We would also like to thank you again for your feedback and tell you that we are open to further improvement through your suggestions.
Kind regards,
The Authors.
Round 2
Reviewer 2 Report
The article is based mainly on information from experts in the aspect of cybersecurity. Despite some reservations, I accept it for publication.